# Differences in Non-Pathogenic Lung-Colonizing Bacteria Among Patients with Different Types of Pneumonia: A Retrospective Study

**DOI:** 10.3390/microorganisms13092099

**Published:** 2025-09-09

**Authors:** Cheng-Yi Hu, Shu-Fang Yao, Yan-Fang Li, Qi-Zhi Wang, Yu-Jun Li, Cheng Sun, Jun Liu, Zhu-Xiang Zhao

**Affiliations:** 1Respiratory Department, Guangzhou First People’s Hospital, School of Medicine, South China University of Technology, Guangzhou 510180, China; 13265516426@163.com (C.-Y.H.); wangqizhi6662010@163.com (Q.-Z.W.); liyujun_0110@163.com (Y.-J.L.); eysunc@scut.edu.cn (C.S.); 18675867725@126.com (J.L.); 2Department of Infectious Diseases, Guangzhou First People’s Hospital, School of Medicine, South China University of Technology, Guangzhou 510180, China; 17819836518@163.com (S.-F.Y.); 13580378865@163.com (Y.-F.L.)

**Keywords:** colonizing bacteria, pneumonia, metagenomic next-generation sequencing, clinical outcomes, lung microbiota, bronchoalveolar lavage, non-pathogenic bacteria, *Rothia mucilaginosa*, *Prevotella melaninogenica*

## Abstract

The clinical impact of non-pathogenic colonizing bacteria in pneumonia remains poorly understood. This retrospective study analyzed the mutual influence of pneumonia and non-pathogenic bacterial flora in the lungs. Bronchoalveolar lavage fluid samples from 483 patients were analyzed using metagenomic next-generation sequencing, and differences in colonizing bacteria in different pneumonia types and their impact on disease prognosis were determined. Patients with bacterial pneumonia exhibited higher and lower colonization rates of *Granulicatella adiacens* and *Streptococcus parasanguinis*, respectively, than those without. Fungal pneumonia showed lower and higher colonization rates of *Abiotrophia defectiva* and *Veillonella parvula*, respectively; viral pneumonia showed higher colonization rates of *Abiotrophia defectiva* and *Streptococcus mitis*. *Rothia mucilaginosa* was associated with shorter duration of fever, and lower risks of sepsis and multiple organ dysfunction syndrome (MODS). *Prevotella melaninogenica* was associated with lower risks of sepsis and MODS. These findings suggest that select non-pathogenic bacteria might influence disease severity and also highlight the need for further investigation into microbiome-based therapeutic strategies, potentially guiding personalized pneumonia treatments.

## 1. Introduction

Pneumonia contributes significantly to the global disease burden, with an annual increase in hospitalization rates and intensive care unit (ICU) admissions. Approximately 649 out of every 100,000 people are hospitalized because of pneumonia, with a mortality rate of approximately 6.5% during hospitalization [1,2]. Identification of the causative pathogen is crucial for effective pneumonia treatment; however, in many hospitalized patients, the pathogen remains unidentified [3]. Conventional microbiological tests often have limitations in terms of sensitivity, timeliness, and the range of detectable pathogens. Metagenomic next-generation sequencing (mNGS) is a valuable tool for diagnosing infections of unknown causes [4,5]. mNGS has shown promise in guiding antibiotic treatment decisions by providing comprehensive and rapid identification of pathogens [6], thus facilitating targeted treatment strategies and potentially reducing the unnecessary use of broad-spectrum antibiotics.

Numerous studies have obtained bronchoalveolar lavage fluid (BALF) through bronchoscopy and confirmed, through pathogen detection, the presence of large numbers of colonizing bacteria in the lungs [7,8,9]. These colonizing bacteria include pathogenic and opportunistic pathogenic microbes and symbiotic communities [10,11]. Non-pathogenic bacterial flora is associated with the prognosis of many diseases, such as chronic obstructive pulmonary disease (COPD), idiopathic pulmonary fibrosis, asthma, and acute respiratory distress syndrome. Furthermore, non-pathogenic bacterial flora influences lung immune responses, including the levels of cytokines such as interleukin (IL)-1, IL-6, and tumor necrosis factor (TNF)-α [12,13]. Determining whether pathological changes in lung structure cause microbial dysbiosis or whether dysbiosis in the microbial community causes structural changes in the lung remains a challenge in lung disease research. The ecological imbalance caused by non-pathogenic bacterial flora may play a pathogenic role in lung disease by modulating the immune microenvironment, including cytokine regulation. However, although previous studies have described the influence of the lung microbiota in chronic respiratory conditions, the relationship between non-pathogenic lung-colonizing bacteria and pneumonia remains unclear. In the present study, we aimed to investigate the interactions and mutual influences between pneumonia and non-pathogenic lung-colonizing bacteria and elucidate the ecological microenvironment of the lower respiratory tract microbiota. The findings of this study can assist in the development of future clinical treatment strategies for pneumonia, can provide new insights into the lung microbial ecology in pneumonia, and may inform microbiome-targeted therapeutic approaches for improving patient outcomes.

## 2. Materials and Methods

### 2.1. Ethics Approval and Consent to Participate

All procedures conducted in this study adhered to the ethical standards set by the appropriate national and institutional committees on human experimentation, as well as the Helsinki Declaration of 1975 and its 2008 revision. All patients included in this study signed informed consent forms for fiberoptic bronchoscopy and alveolar lavage procedures. This retrospective analysis was approved by the Medical Ethics Committee of Guangzhou First People’s Hospital (approval number: K-2023-103-02).

### 2.2. Patient Cohort and Study Design

This retrospective, single-center study examined data from 583 patients diagnosed with pneumonia at the Guangzhou First People’s Hospital, China. Patients who were admitted to our hospital for pneumonia, underwent fiber bronchoscopy during hospitalization, and had their BALF tested for pathogens using mNGS from December 2017 to December 2023 were included. The exclusion criteria were as follows: (1) cancer or connective tissue diseases, (2) hematological disorders such as leukemia and lymphoma, and (3) incomplete clinical data. Figure 1 illustrates a flowchart detailing the patient selection process. The final analysis included data from 483 patients with pneumonia. The baseline was defined as the point at the time of patient admission. The endpoint was defined as death or discharge of the patient. The median hospitalization stay was 10 (range: 8–15) days.

Some of the patients had mixed pneumonia. In addition, some patients had unexplained pneumonia.

### 2.3. Definitions

Unexplained pneumonia was defined as pneumonia for which the specific type could not be determined despite comprehensive testing that included biological evidence, such as sputum culture, blood culture, and BALF-mNGS [14,15]. The pathogenic bacteria that cause pneumonia in patients were identified by two deputy chief physicians with a background in infectious diseases based on the patient’s symptoms and signs, imaging results, bacterial culture results of alveolar lavage fluid, and metagenomic next-generation sequencing results [16,17]. Non-pathogenic colonizing bacteria were defined as bacteria detected by BALF-mNGS that were not considered pathogenic bacteria. Non-pathogenic colonizing bacteria included opportunistic pathogens and commensal bacteria.

### 2.4. Collection of BALF

The specific location of pneumonia was determined using computed tomography. A fiberoptic bronchoscope was inserted into the affected segmental or subsegmental bronchus, and 20 mL of sterile saline (0.9% NaCl at room temperature) was instilled, immediately aspirated, and recovered. All patients underwent fiberoptic bronchoscopy and BALF collection within 1 week of hospitalization.

### 2.5. Cytokine Level Measurement

According to the manufacturer’s instructions, we measured the levels of cytokines such as IL-2, IL-4, IL-6, IL-10, interferon (IFN)-γ, and TNF-α for all patients using the Human Cytokine Standard 27-plex Assay Panel and the Bio-Plex 200 System (Bio-Rad Laboratories, Hercules, CA, USA).

### 2.6. Nucleic Acid Extraction, Library Preparation, and Sequencing

For the extraction of genomic material, we employed a multi-step procedure. Sample processing began with the isolation of DNA molecules by applying the QIAamp^®^ UCP Pathogen DNA Kit manufactured by Qiagen (Hilden, Germany) in accordance with the supplier’s protocol. To enhance microbial DNA purity, contaminating human genetic material was eliminated through a combination of benzonase enzyme (Qiagen) and the detergent Tween 20 (Sigma-Aldrich, St. Louis, MO, USA), following established methods [18]. In parallel, we extracted viral and microbial RNA components from the specimens utilizing the QIAamp^®^ Viral RNA Kit (Qiagen). The quality of extracted RNA was improved by depleting ribosomal RNA sequences with Illumina’s Ribo-Zero rRNA Removal Kit (San Diego, CA, USA) [19]. Subsequently, complementary DNA was synthesized from the purified RNA through an enzymatic reaction with reverse transcriptase and deoxynucleoside triphosphates supplied by Thermo Fisher Scientific (Waltham, MA, USA). We prepared genomic and complementary DNA library constructs utilizing Nextera XT DNA Library Preparation system (Illumina, San Diego, CA, USA) [20]. The pooled libraries underwent sequencing on an NextSeq 550Dx platform (Illumina, San Diego, CA, USA) utilizing a 75-cycle single-end approach, which yielded an average of 20 million sequence reads for each individual library. To establish appropriate controls, we utilized peripheral blood mononuclear cells obtained from healthy individuals, prepared at a density of 10^5^ cells/mL, which were subjected to identical processing procedures alongside each experimental batch. Additionally, we incorporated sterile deionized water as a non-template control (NTC) that underwent the same extraction workflow as the test specimens [21].

### 2.7. Bioinformatic Analyses

Sequence quality filtration was accomplished using the Trimmomatic (v0.39), which facilitated the elimination of poor-quality sequences, adapter-derived contamination, redundant reads, and any fragments not meeting the minimum length threshold of 50 base pairs [22]. To further enhance data quality, we used Kcomplexity with its default settings to filter out sequences characterized by low nucleotide complexity [23]. Sequences originating from the human host were computationally distinguished and subsequently filtered out by conducting alignment against the hg38 human reference assembly utilizing the alignment algorithm implemented in Burrows–Wheeler Aligner (v0.7.17) [24]. The sequence data analysis involved mapping microbial sequence fragments to reference databases through SNAP software (version 1.0beta.18) [25]. For quantitative assessment, we calculated a normalized abundance metric by dividing each microorganism’s reads per million (RPM) value in patient samples by its corresponding value in negative controls, yielding what we termed the RPM-ratio (RPM-r). An organism was confirmed present when this RPM-r exceeded or equaled 5 [19].

### 2.8. Data Collection

We collected comprehensive data for each patient from the medical system, including demographics, comorbidities, symptoms, imaging reports (initial results upon admission), laboratory test results (initial results upon admission), treatments, outcomes, and hospital costs. A dedicated team of physicians and researchers collaborated to cross-check and verify the accuracy of the patient data.

### 2.9. Quantification and Statistical Analysis

For data analysis, we utilized two statistical software packages: IBM’s SPSS, version 25.0 (Armonk, NY, USA), alongside the GraphPad Prism application, version 8.0.0 (San Diego, CA, USA). Upon testing our continuous variables with the Shapiro–Wilk method, we determined that normal distribution patterns were absent across all measurements. Consequently, we have expressed these continuous data as median values accompanied by their interquartile ranges (IQRs). For categorical data, we have opted for numerical counts supplemented with corresponding percentages. When conducting between-group comparisons, non-parametric Mann–Whitney U testing was employed for continuous variables, while either Chi-square analysis or Fisher’s exact probability test was applied to categorical data [26]. The Kaplan–Meier method was used to compare fever duration and hospitalization duration with group comparisons performed using the log-rank test. Spearman’s correlation analysis was used to analyze the correlation between non-pathogenic lung-colonizing bacteria and cytokine levels. The Shannon Diversity Index was calculated using the following formula: $-\sum_{i=1}^{s}\left(p_{i}\log p_{i}\right)$. The Simpson Index was calculated as D = $\sum_{i=1}^{s}p_{i}^{2}$. The Chao Richness Score, used to compute the number of species in a community, was calculated through manual enumeration [27]. Differences in the Shannon Diversity Index and Chao Richness Score among groups were determined using the Mann–Whitney U test or the non-parametric Kruskal–Wallis test. Principal component analysis and principal coordinate analysis were used for analyzing β-species diversity [28,29]. All statistical tests were two-sided, with *p* < 0.05 considered statistically significant.

Through the univariable and multivariable logistic regression analyses, we found that age, sex, comorbidities, antibiotic use, and disease severity may serve as confounding factors affecting patients’ clinical outcomes (Appendix A). To standardize confounding factors and minimize bias, propensity score matching (PSM) analysis was performed. The patients in the groups were matched 1:1 using the greedy-matching nearest-neighbor algorithm with a caliper of 0.02 [30,31]. All confounding factors in this study were balanced between the groups after PSM analysis.

## 3. Results

### 3.1. Baseline Characteristics of All Patients

Appendix A shows the baseline clinical characteristics of all patients in this study. The median age of the patients was 63 years (IQR: 50–71 years), with male patients accounting for 61.7% of the cohort. In total, 26.3%, 18.6%, 7.5%, and 13.0% of the patients had hypertension, diabetes, CHD, and COPD, respectively. Elevated white blood cell counts were observed in 24.0% of the patients. Most patients showed no obvious abnormalities in liver or kidney function or myocardial enzyme levels. In addition, IL-2, IL-4, IL-6, IL-10, TNF-α, and IFN-γ levels were elevated by varying amounts depending on the type of pneumonia. During hospitalization, each patient received a median of 2 antibiotic types (IQR: 1–4) for a median of 9 days (IQR: 7–13 days). In addition, 65.4% of the patients required oxygen inhalation therapy, and 14.3% required ventilator-assisted ventilation. With regard to the disease prognosis, 15.1% of the patients developed severe pneumonia, 10.8% were admitted to the ICU for monitoring, 10.4% experienced multiple organ dysfunction syndrome (MODS), 6.4% developed sepsis, and 3.3% died. The common pathogenic bacteria in bacterial pneumonia patients in this study were *Streptococcus pneumoniae*, *Pseudomonas aeruginosa*, *Haemophilus influenzae*, *Klebsiella pneumoniae*, and *Acinetobacter baumannii* (Appendix A). The common pathogenic fungi in fungal pneumonia patients in this study were *Candida*, *Pneumocystis jirovecii*, and *Aspergillus* (Appendix A).

Appendix A illustrates the characteristics of the non-pathogenic colonizing flora in all patients. Appendix A details the top 15 non-pathogenic colonizing Gram-positive bacteria by colonization rate, with the top three being *Rothia mucilaginosa* (40.4%), *Granulicatella adiacens* (15.9%), and *Abiotrophia defectiva* (11.8%). Appendix A shows the sequences of the top 15 non-pathogenic colonizing Gram-positive bacteria, most of which had quartiles between 10 and 1000. Appendix A displays the top 15 non-pathogenic colonizing Gram-negative bacteria by colonization rate, and the top three were *Veillonella parvula* (34.2%), *Prevotella melaninogenica* (33.1%), and *Veillonella dispar* (14.7%). Appendix A presents the sequences of the top 15 non-pathogenic colonizing Gram-negative bacteria.

### 3.2. Differences in Non-Pathogenic Bacterial Flora Colonizing the Lungs Between Patients with and Those Without Bacterial Pneumonia

Before PSM, compared with patients without bacterial pneumonia, those with bacterial pneumonia exhibited higher colonization rates for *Granulicatella adiacens* (19.6% vs. 8.6%, *p* = 0.002), *Gemella haemolysans* (10.9% vs. 4.3%, *p* = 0.015), *Fusobacterium nucleatum* (16.8% vs. 8.6%, *p* = 0.015), *Porphyromonas gingivalis* (17.1% vs. 5.6%, *p* < 0.001), *Tannerella forsythia* (13.1% vs. 6.8%, *p* = 0.037), *Campylobacter concisus* (12.5% vs. 5.6%, *p* = 0.018), and *Veillonella atypica* (11.5% vs. 4.9%, *p* = 0.019). Furthermore, they showed higher numbers of sequences for *Abiotrophia defectiva* (*p* = 0.040), *Megasphaera micronuciformis* (*p* = 0.044), *Staphylococcus epidermidis* (*p* = 0.027), and *Tannerella forsythia* (*p* = 0.014) (Appendix A).

Considering that confounding factors such as age, sex, and comorbidities might cause differences in the bacterial colonization rate, we performed PSM (Appendix A). After PSM, no statistically significant differences were observed in age, sex, hypertension, diabetes, CHD, or COPD between patients with and those without bacterial pneumonia.

Figure 2 shows the full comparison of colonization rates of patients after PSM. Compared with patients without bacterial pneumonia, those with bacterial pneumonia demonstrated higher colonization rates for *Granulicatella adiacens* (17.5% vs. 7.8%, *p* = 0.010) (Figure 2a), *Gemella haemolysans* (14.3% vs. 3.9%, *p* = 0.002), *Fusobacterium nucleatum* (20.8% vs. 9.1%, *p* = 0.004) (Figure 2c), *Porphyromonas gingivalis* (12.3% vs. 5.8%, *p* = 0.047), *Tannerella forsythia* (14.3% vs. 7.1%, *p* = 0.043), and *Campylobacter concisus* (12.3% vs. 5.8%, *p* = 0.047), along with a lower colonization rate for *Streptococcus parasanguinis* (6.5% vs. 14.3%, *p* = 0.025). Moreover, patients with bacterial pneumonia after PSM had more *Veillonella atypica* sequences (*p* = 0.044) (Figure 2d).

### 3.3. Differences in Non-Pathogenic Bacterial Flora Colonizing the Lungs Between Patients with and Those Without Fungal Pneumonia

Before PSM, compared with patients without fungal pneumonia, those with fungal pneumonia exhibited lower colonization rates of *Abiotrophia defectiva* (5.8% vs. 13.8%, *p* = 0.018), *Solobacterium moorei* (3.3% vs. 8.8%, *p* = 0.045), *Filifactor alocis* (1.7% vs. 8.0%, *p* = 0.013), *Veillonella dispar* (9.1% vs. 16.6%, *p* = 0.044), *Porphyromonas gingivalis* (6.6% vs. 15.5%, *p* = 0.013), and *Tannerella forsythia* (5.8% vs. 12.7%, *p* = 0.035). Conversely, they had higher colonization rates of *Streptococcus mitis* (13.2% vs. 4.7%, *p* = 0.001) and *Veillonella parvula* (47.1% vs. 29.8%, *p* < 0.001). Furthermore, patients with fungal pneumonia had a higher number of sequences for *Rothia mucilaginosa* (*p* = 0.022) and fewer sequences for *Filifactor alocis* (*p* = 0.029) (Appendix A).

Appendix A shows the results of comparisons between patients with and those without fungal pneumonia after PSM. Age, sex, and comorbidities were balanced between the two groups after PSM. Compared with patients without fungal pneumonia, those with fungal pneumonia showed a lower lymphocyte count (*p* < 0.001), a higher neutrophil count (*p* = 0.005), and higher serum creatinine levels (*p* = 0.014). Moreover, these patients were more likely to experience abnormal liver function (aspartate transaminase > 40 U/L) (*p* = 0.034), hypoproteinemia (*p* = 0.001), and myocardial damage (*p* = 0.023), with lower blood levels of IL-10 (*p* = 0.009), TNF-α (*p* = 0.004), and IFN-γ (*p* = 0.008).

After PSM, compared with patients without fungal pneumonia, those with fungal pneumonia showed lower colonization rates of *Abiotrophia defectiva* (5.9% vs. 17.8%, *p* = 0.005) (Figure 2e) and *Porphyromonas gingivalis* (6.8% vs. 16.1%, *p* = 0.024) (Figure 2g) and higher colonization rates of *Veillonella parvula* (48.3% vs. 31.4%, *p* = 0.008). Furthermore, they exhibited lower numbers of sequences for *Filifactor alocis* (*p* = 0.037) (Figure 2f) and *Fusobacterium nucleatum* (*p* = 0.012) (Figure 2h).

### 3.4. Differences in Non-Pathogenic Bacterial Flora Colonizing the Lungs Between Patients with and Those Without Viral Pneumonia

Before PSM, compared with patients without viral pneumonia, those with viral pneumonia exhibited lower colonization rates of *Solobacterium moorei* (4.2% vs. 9.2%, *p* = 0.045) and higher colonization rates of *Streptococcus infantis* (14.3% vs. 7.9%, *p* = 0.028) and *Streptococcus mitis* (13.1% vs. 3.5%, *p* < 0.001). Furthermore, patients with viral pneumonia had higher numbers of sequences of *Rothia mucilaginosa* (*p* < 0.001), *Granulicatella adiacens* (*p* = 0.010), *Streptococcus oralis* (*p* = 0.003), *Staphylococcus epidermidis* (*p* = 0.024), *Streptococcus infantis* (*p* < 0.001), *Streptococcus mitis* (*p* = 0.032), *Veillonella parvula* (*p* = 0.013), and *Capnocytophaga granulosa* (*p* = 0.036) (Appendix A).

Appendix A shows the clinical characteristics of patients with and those without viral pneumonia after PSM. Compared with patients without viral pneumonia after PSM, those with viral pneumonia were more likely to exhibit hypoproteinemia (*p* = 0.013) and had lower lymphocyte counts (*p* = 0.003) and blood IFN-γ levels (*p* = 0.019).

After PSM, compared with patients without viral pneumonia, those with viral pneumonia had higher colonization rates of *Abiotrophia defectiva* (15.0% vs. 6.9%, *p* = 0.020) (Figure 2i), *Streptococcus infantis* (14.4% vs. 7.5%, *p* = 0.049), and *Streptococcus mitis* (13.1% vs. 2.5%, *p* < 0.001). Furthermore, patients with viral pneumonia after PSM had a higher number of sequences for *Rothia mucilaginosa* (*p* < 0.001) (Figure 2j), *Granulicatella adiacens* (*p* = 0.044), *Streptococcus oralis* (*p* = 0.004), *Staphylococcus epidermidis* (*p* = 0.041), *Streptococcus infantis* (*p* = 0.015), and *Capnocytophaga granulosa* (*p* = 0.014) (Figure 2l).

### 3.5. Impact of Rothia Mucilaginosa Colonization in Patients with Pneumonia

We conducted PSM to account for confounding factors. After PSM, confounding factors such as age, sex, comorbidities, antibiotic use, pneumonia type, and disease severity were balanced (Table 1). *Rothia mucilaginosa* colonization did not affect laboratory results, including white blood cell counts, lymphocyte levels, liver and kidney function, myocardial enzyme levels, and cytokine concentrations.

Before PSM, lung colonization by *Rothia mucilaginosa* was associated with shorter hospital stays (*p* = 0.006) (Appendix A). After PSM, patients with pneumonia with *Rothia mucilaginosa* colonization exhibited shorter duration of fever (*p* = 0.015), and lower risks of sepsis (*p* = 0.032) and MODS (*p* = 0.034) than patients without *Rothia mucilaginosa* colonization (Figure 3).

### 3.6. Impact of Veillonella Parvula Colonization in Patients with Pneumonia

Before PSM, significant differences were observed in age, comorbidities, and pneumonia types between patients with pneumonia and those without *Veillonella parvula* colonization (Table 2). Consequently, we conducted PSM to control for these confounding factors and analyze the impact of *Veillonella parvula* on pneumonia outcomes. After PSM, we found that *Veillonella parvula* colonization did not affect laboratory results, including white blood cell counts, lymphocyte levels, procalcitonin levels, liver and kidney function, cardiac enzymes, and cytokine levels.

Before PSM, *Veillonella parvula* colonization was associated with higher hospital costs (*p* = 0.036) and greater reliance on supplemental oxygen (*p* = 0.002) (Appendix A). After PSM, the proportion of patients with pneumonia requiring oxygen inhalation therapy remained higher among those with *Veillonella parvula* colonization (*p* < 0.001) (Figure 4).

### 3.7. Impact of Prevotella Melaninogenica Colonization in Patients with Pneumonia

Table 3 outlines the clinical features of patients with pneumonia with *Prevotella melaninogenica* colonization. After PSM, higher levels of IL-4 (0.35 [0.02–0.69] vs. 0.21 [0.01–0.57], *p* = 0.014) were observed in patients with *Prevotella melaninogenica* colonization.

Before PSM, *Prevotella melaninogenica* colonization in patients with pneumonia was associated with a reduced need for oxygen inhalation therapy (*p* = 0.002) (Appendix A) and ventilator-assisted therapy (*p* = 0.014), lower risks of progression to sepsis (*p* = 0.013) and MODS (*p* = 0.007), lower likelihood of ICU transfer (*p* = 0.024), shorter hospital stays (*p* < 0.001), and lower hospital costs (*p* < 0.001). After PSM, *Prevotella melaninogenica* colonization reduced the risks of sepsis (*p* = 0.032) and MODS (*p* = 0.010) (Figure 5).

### 3.8. Correlations Among Non-Pathogenic Colonizing Bacteria and Cytokine Levels

Figure 6 suggests a potential correlation between colonizing bacteria and cytokine levels.

In patients with bacterial pneumonia (Figure 6a), positive correlations were found between IL-6 and *Veillonella parvula* (*p* < 0.001), while TNF-α positively correlated with *Capnocytophaga granulosa* (*p* = 0.009). A negative correlation was observed between IL-10 and *Neisseria elongate* (*p* = 0.019). In patients with fungal pneumonia (Figure 6b), positive correlations were found between TNF-α and *Fusobacterium nucleatum* (*p* < 0.001). A negative correlation was observed between IL-10 and *Rothia mucilaginosa* (*p* = 0.004). In patients with viral pneumonia (Figure 6c), positive correlations were found between TNF-α and *Fusobacterium nucleatum* (*p* = 0.005), while IL-2 positively correlated with *Capnocytophaga granulosa* (*p* = 0.011).

### 3.9. Comparison of Microbial Diversity of Non-Pathogenic Lung-Colonizing Bacteria in Patients with Different Types of Pneumonia

Figure 7a shows the relative abundances of colonizing bacteria in patients with different types of pneumonia. Indices measuring microbial diversity, including Shannon, Simpson, InvSimpson, richness, the abundance-based coverage estimator (ACE), and Chao1, were similar among the groups at the genus and species levels (Figure 7b) (*p* > 0.05). β-diversity was also not significantly different among the groups (Figure 7c).

## 4. Discussion

This study aimed to investigate the role of non-pathogenic lung-colonizing bacteria in pneumonia prognosis using mNGS analysis of BALF samples. Our study identified the top 15 g-positive and Gram-negative bacteria with the highest colonization rates. We found that different types of pneumonia led to variations in the bacterial communities colonizing the lungs. As previously reported, sarcoidosis affects the microbiota in the lower respiratory tract [32]. Asthma and COPD may also influence the respiratory microbiome [33]. Certain colonizing bacteria may thrive in certain respiratory diseases, resulting in their widespread colonization and dominance within the respiratory tract of patients with such diseases, leading to higher colonization rates and increased presence of these bacteria within the entire microbial community.

Furthermore, we found that the bacteria colonizing the lungs affected the prognosis of pneumonia. In this study, the bacteria with the highest colonization rates were *Rothia mucilaginosa* (40.4%), *Veillonella parvula* (34.2%), and *Prevotella melaninogenica* (33.1%). Therefore, we focused our analysis on these three bacteria, with *Rothia mucilaginosa* representing Gram-positive bacteria and *Veillonella parvula* and *Prevotella melaninogenica* representing Gram-negative bacteria. Previous studies have indicated the presence of a core lung bacterial microbiome, comprising *Prevotella*, *Veillonella*, and *Rothia* [34,35]. For each individual, the composition of the pulmonary microbiome is based on the balance between microbial transmission from the upper respiratory tract and microbial elimination by the host’s defense mechanisms [36,37]. When the colonizing bacteria that dominate the colonizing flora differ, differences in the prognosis of lung diseases may arise [38]. Despite numerous previous studies reporting the negative impact of intrapulmonary colonizing bacteria on pulmonary diseases, our research revealed that Rothia mucilaginosa and Prevotella melaninogenica might be beneficial for the prognosis of pneumonia patients. *Rothia mucilaginosa* was associated with shorter duration of fever, and lower risks of sepsis and MODS. *Prevotella melaninogenica* was associated with lower risks of sepsis and MODS. Beyond pneumonia, colonizing bacteria influence the occurrence and development of lung cancer and sarcoidosis [39]. Furthermore, lung function correlates significantly with the richness and diversity of colonizing bacteria [40]. However, some studies have reported that colonizing bacteria do not affect the prognosis of certain lung diseases; for example, short-term changes in the composition of the airway microbiota have not been associated with the deterioration of lung disease in cystic fibrosis [41]. *Veillonella* and *Prevotella* play crucial roles in respiratory system health as unique components of the normal lung microbiota and other moist epithelial cell populations. Notably, a negative correlation has been observed between influenza-specific H1 IgA titers and *Veillonella* [42]. Furthermore, *Prevotella* is more common in healthy individuals than in children with asthma and patients with COPD [43].

Our findings suggest that intrapulmonary colonizing bacteria may affect cytokine levels. A previous study analyzed IL-1β levels and airway microbiota in patients with bronchiectasis, finding a correlation between the relative abundance of Proteobacteria and IL-1β [44]. Furthermore, the abundance of resident bacteria may be associated with airway immunological characteristics that primarily involve a reduction in TNF-α and IL-1β levels [45]. In addition, the α-diversity of the lung microbiota in patients with lung transplants may be associated with IL-2 levels in BALF.

### Limitations of the Study

This study has some limitations. The timing of fiberoptic bronchoscopy was inconsistent across all patients, potentially leading to differences in the number and types of colonizing bacteria detected using mNGS. At the time of BALF acquisition, the position of the bronchoscope was not always consistent during bronchoscopy, potentially resulting in differences in the types of colonizing bacteria detected. Despite our quality control monitoring, the sampling environment, detection environment, and pollutants may have still affected the results of bacterial colonization detected by mNGS. The detected colonizing bacteria may comprise PPMs, symbiotic bacteria, and other potential sources of contamination. Although clinicians recognize the presence of normal respiratory microbiota within the lungs, the results of mNGS should be interpreted with caution. Quantitative methods and interpretation standards for mNGS are needed to distinguish pathogens from commensals and minimize false-positive results. In addition, since our research population only came from China, it is difficult to generalize the conclusions to other races or countries. Furthermore, antibiotic use in patients with pneumonia during hospitalization may affect the number and types of colonizing bacteria. In this study, non-pathogenic bacteria include opportunistic pathogens and commensal bacteria, and opportunistic pathogens may affect clinical outcomes in patients with weakened immunity. Finally, this retrospective study was conducted at a single center with a limited patient sample. These limitations highlight the need for standardized protocols, and prospective multicenter clinical studies are needed to validate our findings and fully elucidate the clinical implications of lung-colonizing microbiota in pneumonia management.

In summary, our findings support the hypothesis that non-pathogenic colonizing bacteria may influence pneumonia outcomes through immune modulation. Future studies should aim to explore causal mechanisms and evaluate the potential of microbiome-targeted therapies in respiratory infections.

## 5. Conclusions

The bacterial communities colonizing the lungs of patients with different types of pneumonia are distinct. The presence of certain colonizing bacteria, such as *Rothia mucilaginosa* and *Prevotella melaninogenica*, may improve patient prognosis. Our findings highlight the potential value of characterizing non-pathogenic lung-colonizing bacteria as part of the diagnostic and prognostic assessment in pneumonia management. Future research should focus on exploring the mechanistic role of these commensal bacteria in immune modulation and disease progression, as well as their possible therapeutic implications.

## Figures and Tables

**Figure 1 microorganisms-13-02099-f001:**
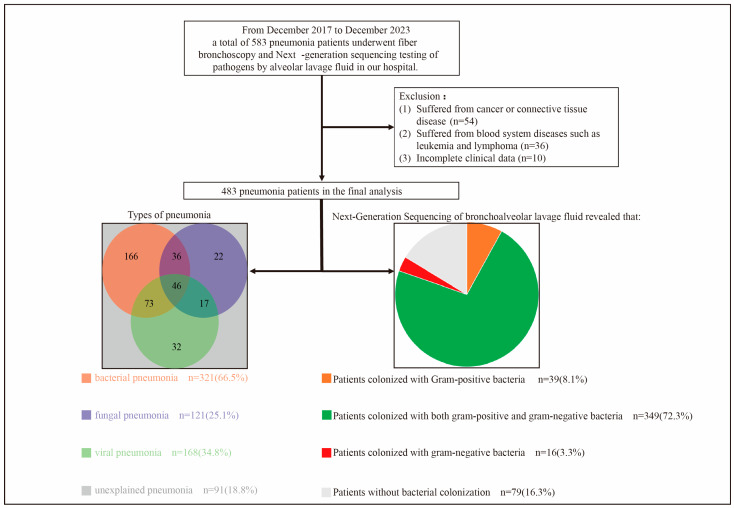
Flowchart of patient selection.

**Figure 2 microorganisms-13-02099-f002:**
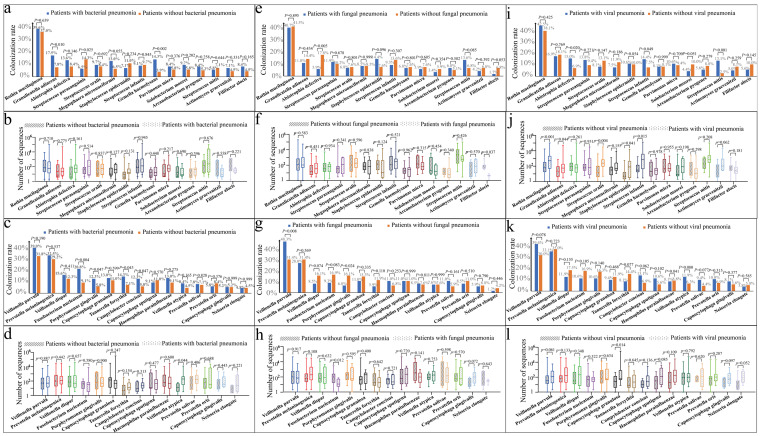
Comparison of colonization rates with sequence numbers of Gram-positive and Gram-negative bacteria in the lungs of patients with different types of pneumonia after propensity score matching (PSM). (**a**) Comparison of the colonization rate of Gram-positive bacteria between patients with and without bacterial pneumonia. (**b**) Comparison of the number of sequenced Gram-positive bacteria between patients with and those without bacterial pneumonia (**c**). Comparison of the colonization rate of Gram-negative bacteria between patients with and those without bacterial pneumonia. (**d**) Comparison of the number of sequenced Gram-negative bacteria between patients with and those without bacterial pneumonia. (**e**) Comparison of the colonization rate of Gram-positive bacteria between patients with and those without fungal pneumonia. (**f**) Comparison of the number of sequenced Gram-positive bacteria between patients with and those without fungal pneumonia. (**g**) Comparison of the colonization rate of Gram-negative bacteria between patients with and those without fungal pneumonia. (**h**) Comparison of the number of sequenced Gram-negative bacteria between patients with and those without fungal pneumonia. (**i**) Comparison of the colonization rate of Gram-positive bacteria between patients with and those without viral pneumonia. (**j**) Comparison of the number of sequenced Gram-positive bacteria between patients with and those without viral pneumonia. (**k**) Comparison of the colonization rate of Gram-negative bacteria between patients with and those without viral pneumonia. (**l**) Comparison of the number of sequenced Gram-negative bacteria between patients with and those without viral pneumonia.

**Figure 3 microorganisms-13-02099-f003:**
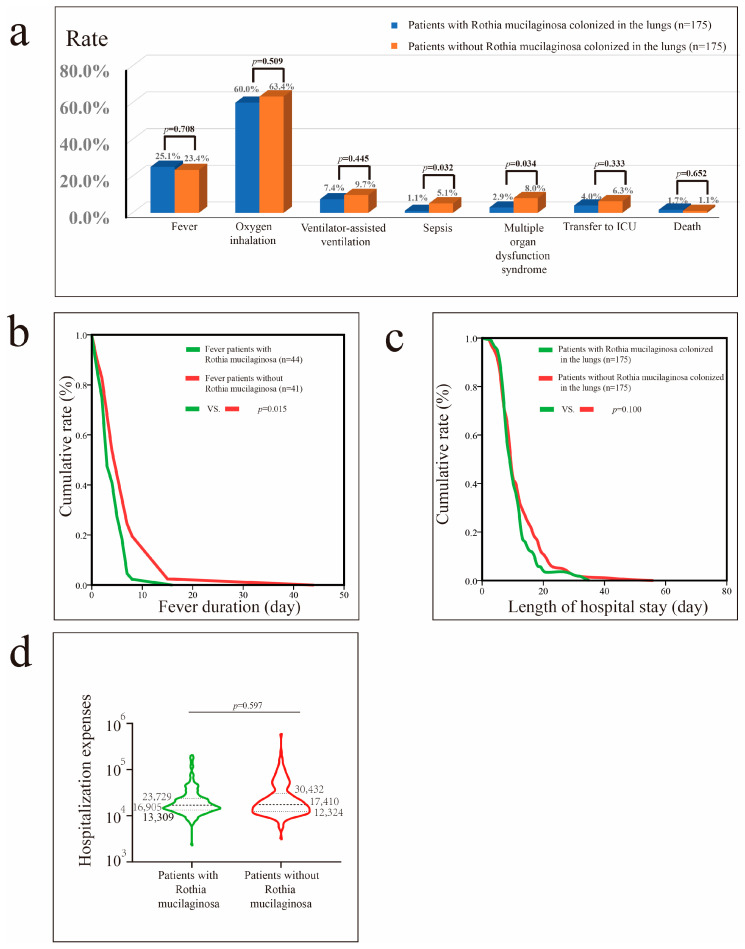
Impact of *Rothia mucilaginosa* colonization in patients with pneumonia after propensity score matching (PSM). (**a**) Impact of *Rothia mucilaginosa* colonization on fever, oxygen inhalation, ventilator-assisted ventilation, sepsis, MODS, transfer to the ICU, and mortality rates. (**b**) Impact of *Rothia mucilaginosa* colonization on fever duration in patients with fever. (**c**) Impact of *Rothia mucilaginosa* colonization on hospital stay in patients with pneumonia. (**d**) Impact of *Rothia mucilaginosa* colonization on hospital costs in patients with pneumonia.

**Figure 4 microorganisms-13-02099-f004:**
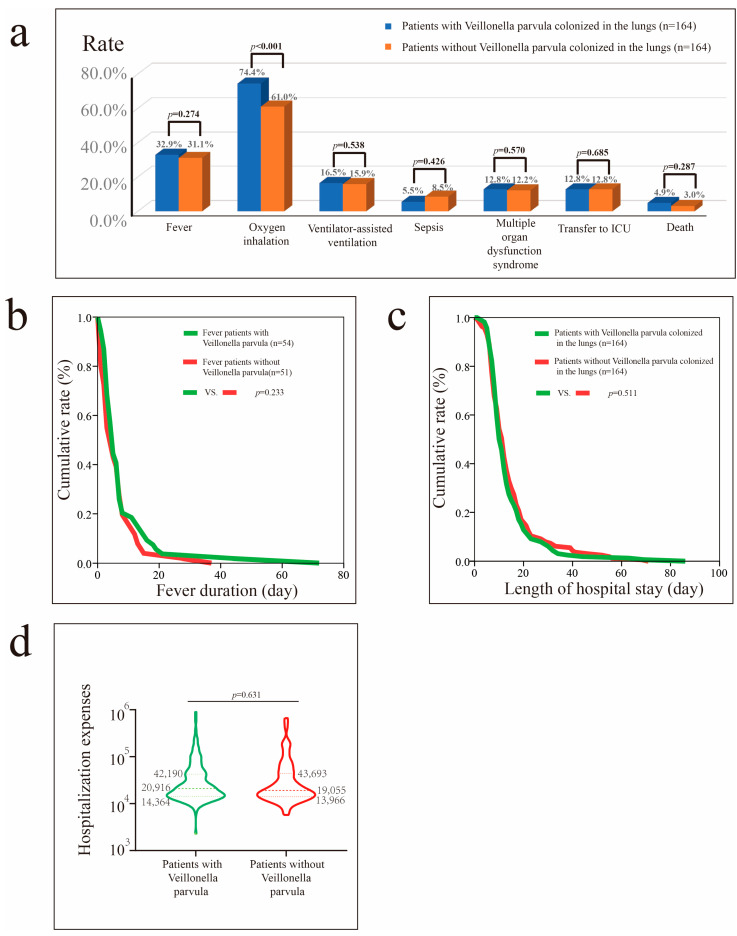
Impact of *Veillonella parvula* colonization in patients with pneumonia after PSM. (**a**) Impact of *Veillonella parvula* colonization on fever, oxygen inhalation, ventilator-assisted ventilation, sepsis, MODS, transfer to the ICU, and mortality rate. (**b**) Impact of *Veillonella parvula* on fever duration in patients with fever. (**c**) Impact of *Veillonella parvula* on hospital stay in patients with pneumonia. (**d**) Impact of *Veillonella parvula* on hospital costs in patients with pneumonia.

**Figure 5 microorganisms-13-02099-f005:**
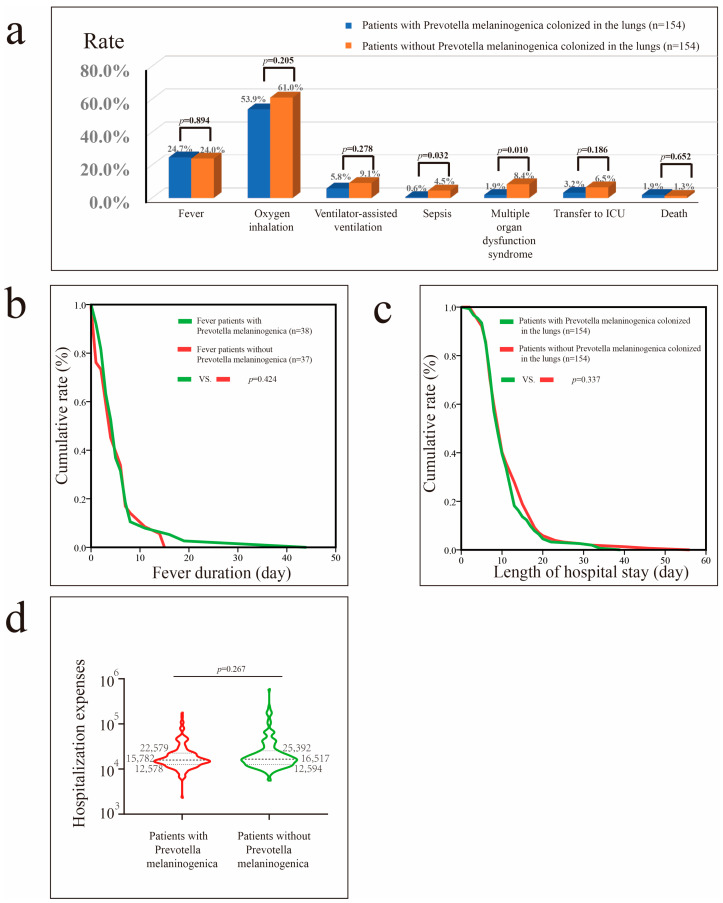
Impact of *Prevotella melaninogenica* colonization in patients with pneumonia after PSM. (**a**) Impact of *Prevotella melaninogenica* colonization on fever, oxygen inhalation, ventilator-assisted ventilation, sepsis, MODS, transfer to ICU, and mortality rates. (**b**) Impact of *Prevotella melaninogenica* on fever duration in patients with fever. (**c**) Impact of *Prevotella melaninogenica* on hospital stay in patients with pneumonia. (**d**) Impact of *Prevotella melaninogenica* on hospital costs in patients with pneumonia.

**Figure 6 microorganisms-13-02099-f006:**
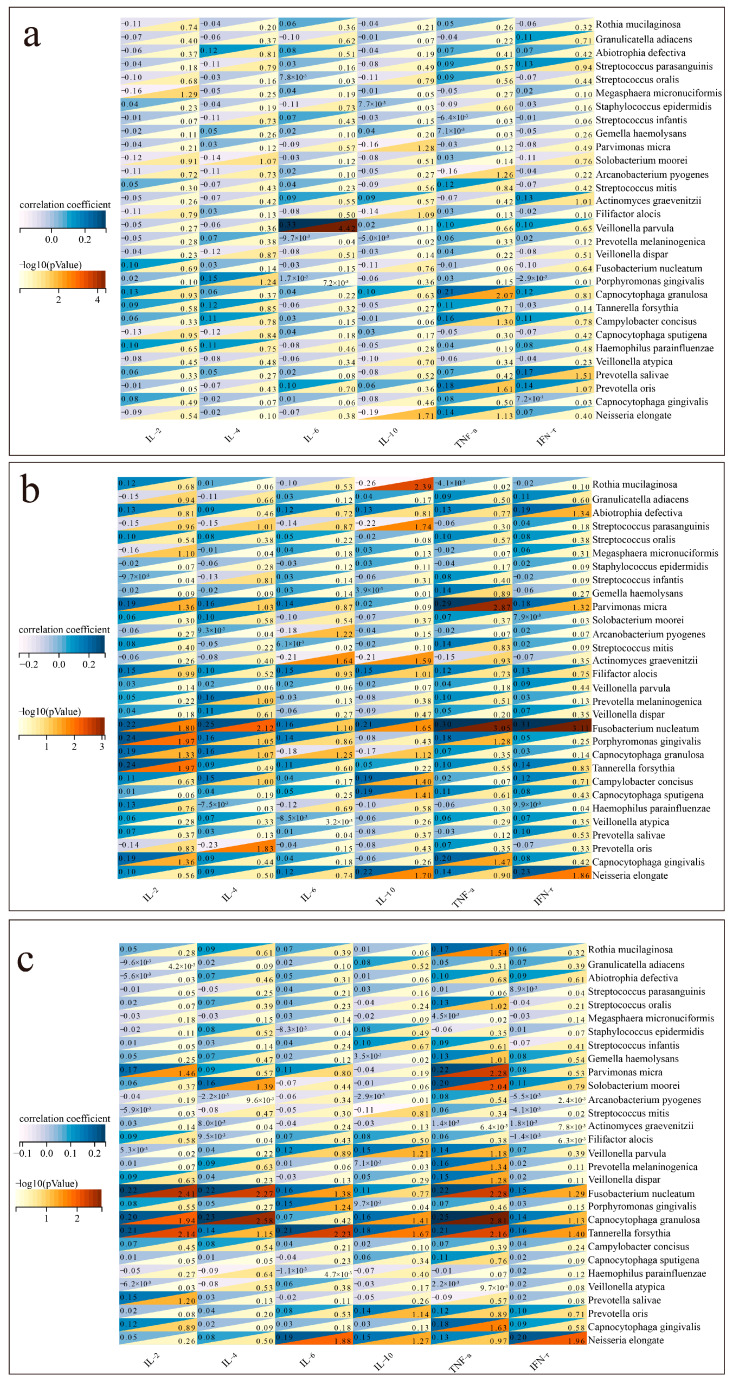
Correlation between colonizing bacteria and cytokines. (**a**) Correlation between colonizing bacteria and cytokines in patients with bacterial pneumonia. (**b**) Correlation between colonizing bacteria and cytokines in patients with fungal pneumonia. (**c**) Correlation between colonizing bacteria and cytokines in patients with viral pneumonia.

**Figure 7 microorganisms-13-02099-f007:**
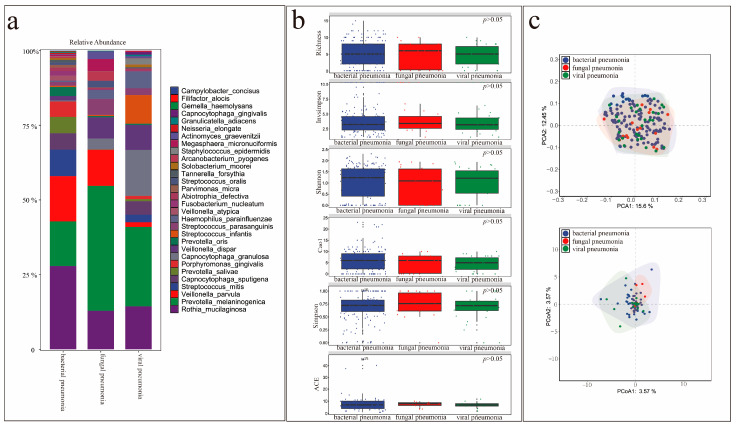
Comparison of microbial diversity of non-pathogenic colonizing bacteria in the lungs of patients with different types of pneumonia. (**a**) Abundance of the top 30 colonizing bacteria in patients with different types of pneumonia. (**b**) Comparison of α-species diversity in patients with different types of pneumonia. (**c**) Comparison of β-species diversity in patients with different types of pneumonia.

**Table 1 microorganisms-13-02099-t001:** Clinical characteristics of pneumonia patients with or without *Rothia mucilaginosa* colonization before and after PSM.

	Before PSM			After PSM		
Characteristics	Patients with *Rothia mucilaginosa* Colonization (*n* = 195)	Patients Without *Rothia mucilaginosa* Colonization (*n* = 288)	*p*-Value	Patients with *Rothia mucilaginosa* Colonization (*n* = 175)	Patients Without *Rothia mucilaginosa* Colonization (*n* = 175)	Adjusted *p*-Value
**Age, years**	63 (51–70)	63 (48–71)	0.945	63 (50–69)	61 (48–71)	0.894
**Sex (men)**	133 (68.2%)	165 (57.3%)	0.015	116 (66.3%)	114 (65.1%)	0.822
**Comorbidities**						
Hypertension	44 (22.6%)	83 (28.8%)	0.125	38 (21.7%)	38 (21.7%)	>0.999
Diabetes	32 (16.4%)	58 (20.1%)	<0.001	28 (16.0%)	27 (15.4%)	0.883
CHD	10 (5.1%)	26 (9.0%)	0.109	7 (4.0%)	5 (2.9%)	0.557
COPD	28 (14.4%)	35 (12.2%)	0.480	23 (13.1%)	23 (13.1%)	>0.999
**Pneumonia**						
Bacterial pneumonia	135 (69.2%)	186 (64.6%)	0.288	118 (67.4%)	119 (68.0%)	0.909
Fungal pneumonia	49 (25.1%)	72 (25.0%)	0.975	40 (22.9%)	40 (22.9%)	>0.999
Viral pneumonia	71 (36.4%)	97 (33.7%)	0.537	65 (37.1%)	63 (36.0%)	0.824
**Antibiotic use**						
Cumulative type of antibiotic use	2 (1–3)	2 (1–4)	0.001	2 (1–3)	2 (1–3)	0.615
Cumulative antibiotic use time	8 (6–13)	10 (7–15)	0.022	8 (6–12)	9 (6–13)	0.629
**Disease severity**						
Severe pneumonia	26 (13.3%)	47 (16.3)	0.369	17 (9.7%)	17 (9.7%)	>0.999
**Laboratory results**						
WBC (10^9^/L)	7.81 (6.03–9.53)	7.41 (5.84–10.16)	0.830	7.67 (5.96–9.46)	7.42 (5.82–10.18)	0.786
LY (10^9^/L)	1.43 (0.98–2.00)	1.48 (1.01–2.00)	0.705	1.46 (1.01–2.02)	1.49 (1.05–2.00)	0.931
Mono (10^9^/L)	0.55 (0.42–0.77)	0.53 (0.38–0.74)	0.296	0.54 (0.42–0.76)	0.53 (0.39–0.74)	0.423
NE (10^9^/L)	5.47 (3.73–7.17)	5.21 (3.42–7.91)	0.944	5.28 (3.60–7.00)	5.05 (3.39–7.59)	0.631
TD (μmol/L)	10.4 (8.1–14.1)	10.5 (8.1–13.5)	0.674	10.4 (8.4–13.8)	10.40 (8.10–13.50)	0.725
Cr (μmol/L)	73 (62–90)	69 (57–87)	0.023	73 (62–89)	72 (57–89)	0.299
AST (U/L)	22 (17–26)	21 (17–28)	0.933	21 (17–25)	20 (17–26)	0.720
ALT (U/L)	16 (11–26)	16 (11–27)	0.744	16 (11–25)	16 (10–24)	0.740
DD (μmol/L)	2.0 (1.6–2.8)	2.0 (1.6–2.7)	0.983	2.0 (1.6–2.8)	2.0 (1.5–2.7)	0.924
ID (μmol/L)	8.2 (6.5–11.5)	8.5 (6.5–10.9)	0.681	8.2 (6.6–11.4)	8.5 (6.4–11)	0.868
ALB (g/L)	36.0 (32.5–38.9)	34.8 (30.8–38.7)	0.045	36.3 (33.2–38.9)	35.2 (31.0–38.9)	0.143
CK-MB (ng/mL)	10 (2–15)	10 (3–15)	0.661	10 (3–15)	10 (3–15)	0.887
cTnI-HS (ng/mL)	0.004 (0.003–0.009)	0.006 (0.003–0.011)	0.059	0.004 (0.002–0.008)	0.005 (0.003–0.009)	0.070
IL-2 (pg/mL)	0.34 (0.01–0.96)	0.40 (0.01–1.00)	0.609	0.42 (0.01–1.01)	0.35 (0.01–0.99)	0.786
IL-4 (pg/mL)	0.30 (0.01–0.64)	0.25 (0.01–0.63)	0.594	0.33 (0.01–0.64)	0.26 (0.01–0.64)	0.663
IL-6 (pg/mL)	70.21 (11.88–193.82)	83.80 (13.59–220.89)	0.525	70.21 (11.51–190.29)	93.79 (15.58–257.28)	0.173
IL-10 (pg/mL)	8.51 (2.47–39.36)	9.05 (2.40–35.15)	0.924	7.97 (2.45–39.36)	9.31 (2.31–36.38)	0.855
TNF-α (pg/mL)	1.65 (0.01–6.86)	1.08 (0.01–6.03)	0.435	2.04 (0.01–7.72)	1.81 (0.01–7.07)	0.516
IFN-γ (pg/mL)	1.91 (0.01–6.98)	1.58 (0.01–7.52)	0.886	2.08 (0.01–7.07)	1.85 (0.01–7.36)	0.986

PSM: propensity score matching; CHD: coronary heart disease; COPD: chronic obstructive pulmonary disease; WBC: white blood cell count; LY: lymphocyte count; Mono: monocyte count; NE: neutrophil count; TD: total bilirubin; Cr: serum creatinine; AST: aspartate transaminase; ALT: alanine aminotransferase; DD: direct bilirubin; ID: indirect bilirubin; ALB: albumin; CK-MB: creatine kinase-MB; cTnI-HS: high-sensitivity cardiac troponin I; IL: interleukin; TNF: tumor necrosis factor; IFN: interferon.

**Table 2 microorganisms-13-02099-t002:** Clinical characteristics of pneumonia patients with or without *Veillonella parvula* colonization before and after PSM.

	Before PSM			After PSM		
Characteristics	Patients with *Veillonella parvula* Colonization (*n* = 165)	Patients Without *Veillonella parvula* Colonization (*n* = 318)	*p*-Value	Patients with *Veillonella parvula* Colonization (*n* = 164)	Patients Without *Veillonella parvula* Colonization (*n* = 164)	Adjusted *p*-Value
**Age, years**	65 (54–73)	61 (48–69)	0.005	65 (54–73)	64 (52–71)	0.380
**Sex (men)**	106 (64.2%)	192 (60.4%)	0.407	106 (64.6%)	108 (65.9%)	0.817
**Comorbidities**						
Hypertension	49 (29.7%)	78 (24.5%)	0.221	48 (29.3%)	46 (28.0%)	0.807
Diabetes	40 (24.2%)	50 (15.7%)	0.023	39 (23.8%)	32 (19.5%)	0.348
CHD	17 (10.3%)	19 (6.0%)	0.086	17 (10.4%)	15 (9.1%)	0.710
COPD	23 (13.9%)	40 (12.6%)	0.674	23 (14.0%)	20 (12.2%)	0.624
**Pneumonia**						
Bacterial pneumonia	113 (68.5%)	208 (65.4%)	0.497	112 (68.3%)	119 (72.6%)	0.397
Fungal pneumonia	57 (34.5%)	64 (20.1%)	0.001	56 (34.1%)	54 (32.9%)	0.815
Viral pneumonia	66 (40.0%)	102 (32.1%)	0.083	66 (40.2%)	66 (40.2%)	>0.999
**Antibiotic use**						
Cumulative type of antibiotic use	2 (1–4)	2 (1–4)	0.410	2 (1–4)	2 (1–4)	0.595
Cumulative antibiotic use time	9 (7–14)	9 (6–13)	0.283	9 (7–14)	10 (7–15)	0.657
**Disease severity**						
Severe pneumonia	26 (15.8%)	47 (14.8%)	0.779	26 (15.9%)	31 (18.9%)	0.804
**Laboratory results**						
WBC (10^9^/L)	7.66 (6.15–10.43)	7.47 (5.84–9.74)	0.376	7.66 (6.13–10.47)	7.21 (5.79–9.69)	0.241
LY (10^9^/L)	1.24 (0.89–1.93)	1.56 (1.10–2.02)	0.004	1.24 (0.88–1.92)	1.46 (0.98–1.94)	0.154
Mono (10^9^/L)	0.56 (0.39–0.80)	0.53 (0.40–0.74)	0.372	0.56 (0.39–0.80)	0.53 (0.39–0.74)	0.461
NE (10^9^/L)	5.66 (3.79–7.77)	5.21 (3.40–7.22)	0.136	5.66 (3.78–7.79)	5.05 (3.40–7.15)	0.144
TD (μmol/L)	10.2 (7.4–13.3)	10.5 (8.3–13.8)	0.123	10.2 (7.4–13.3)	10.8 (8.5–14.4)	0.059
Cr (μmol/L)	72 (60–90)	70 (58–86)	0.095	72 (60–90)	72.5 (59.3–89.8)	0.576
AST (U/L)	21 (17–26)	21 (17–28)	0.901	21 (17–26)	22 (18–31)	0.297
ALT (U/L)	15 (10–27)	16 (11–26)	0.299	15 (10–27)	17 (11–33)	0.172
DD (μmol/L)	2.0 (1.6–2.9)	2.0 (1.5–2.7)	0.820	2.0 (1.6–2.9)	2.2 (1.6–3.2)	0.356
ID (μmol/L)	7.9 (5.9–10.7)	8.5 (6.9–11.3)	0.058	8.0 (5.9–10.8)	8.6 (6.8–11.4)	0.073
ALB (g/L)	34.8 (30.5–38.1)	35.3 (31.8–39.0)	0.035	34.8 (30.5–38.1)	34.9 (30.9–38.5)	0.524
CK-MB (ng/mL)	9 (2–16)	10 (3–15)	0.605	10 (2–16)	10 (3–15)	0.926
cTnI-HS (ng/mL)	0.006 (0.003–0.013)	0.005 (0.002–0.009)	0.010	0.006 (0.003–0.013)	0.006 (0.003–0.012)	0.561
IL-2 (pg/mL)	0.34 (0.01–0.92)	0.44 (0.01–1.01)	0.553	0.34 (0.01–0.90)	0.33 (0.01–0.90)	0.892
IL-4 (pg/mL)	0.31 (0.01–0.63)	0.25 (0.01–0.63)	0.766	0.31 (0.01–0.62)	0.22 (0.01–0.60)	0.410
IL-6 (pg/mL)	91.44 (29.27–206.78)	63.48 (9.09–221.41)	0.112	91.30 (29.10–206.54)	64.78 (9.32–241.30)	0.294
IL-10 (pg/mL)	10.97 (3.11–40.57)	8.22 (2.26–35.20)	0.150	10.97 (3.08–40.59)	7.67 (2.33–29.61)	0.171
TNF-α (pg/mL)	1.81 (0.01–6.41)	0.88 (0.01–6.05)	0.675	1.74 (0.01–6.42)	0.64 (0.01–7.74)	0.840
IFN-γ (pg/mL)	2.23 (0.01–7.23)	1.55 (0.01–7.36)	0.392	2.22 (0.01–7.07)	1.48 (0.01–7.01)	0.352

PSM: propensity score matching; CHD: coronary heart disease; COPD: chronic obstructive pulmonary disease; WBC: white blood cell count; LY: lymphocyte count; Mono: monocyte count; NE: neutrophil count; TD: total bilirubin; Cr: serum creatinine; AST: aspartate transaminase; ALT: alanine aminotransferase; DD: direct bilirubin; ID: indirect bilirubin; ALB: albumin; CK-MB: creatine kinase-MB; cTnI-HS: high-sensitivity cardiac troponin I; IL: interleukin; TNF: tumor necrosis factor; IFN: interferon.

**Table 3 microorganisms-13-02099-t003:** Clinical characteristics of pneumonia patients with or without *Prevotella melaninogenica* colonization before and after PSM.

	Before PSM			After PSM		
Characteristics	Patients with *Prevotella melaninogenica* Colonization (*n* = 160)	Patients Without *Prevotella melaninogenica* Colonization (*n* = 323)	*p*-Value	Patients with *Prevotella melaninogenica* Colonization (*n* = 154)	Patients Without *Prevotella melaninogenica* Colonization (*n* = 154)	Adjusted *p*-Value
**Age, years**	60.5 (46–69.75)	63 (51–71)	0.114	60 (46 –69)	60 (46–68)	0.750
**Sex (men)**	100 (62.5%)	198 (61.3%)	0.799	95 (61.7%)	92 (59.7%)	0.726
**Comorbidities**						
Hypertension	42 (26.3%)	85 (26.3%)	0.998	39 (25.3%)	33 (21.4%)	0.419
Diabetes	16 (10.0%)	74 (22.9%)	0.001	15 (9.7%)	7 (4.5%)	0.077
CHD	13 (8.1%)	23 (7.1%)	0.692	11 (7.1%)	8 (5.2%)	0.477
COPD	22 (13.8%)	41 (12.7%)	0.746	22 (14.3%)	20 (13.0%)	0.740
**Pneumonia**						
Bacterial pneumonia	113 (70.6%)	208 (64.4%)	0.172	108 (70.1%)	96 (62.3%)	0.148
Fungal pneumonia	33 (20.6%)	88 (27.2%)	0.114	31 (20.1%)	38 (24.7%)	0.339
Viral pneumonia	55 (34.4%)	113 (35.0%)	0.895	54 (35.1%)	51 (33.1%)	0.718
**Antibiotic use**						
Cumulative type of antibiotic use	2 (1–3)	2 (1–4)	<0.001	2 (1 –3)	2 (1–3)	0.477
Cumulative antibiotic use time	8 (5–12)	10 (7–15)	<0.001	8 (5 –12)	8 (6–12)	0.331
**Disease severity**						
Severe pneumonia	18 (11.3)	55 (17.0)	0.095	13 (8.4%)	16 (10.4%)	0.558
**Laboratory results**						
WBC (10^9^/L)	7.52 (5.85–9.38)	7.59 (5.96–10.09)	0.443	7.52 (5.82 –9.07)	7.12 (5.73–9.46)	0.802
LY (10^9^/L)	1.51 (0.97–2.02)	1.44 (0.99–1.98)	0.720	1.52 (1.04 –2.02)	1.55 (1.13–2.01)	0.752
Mono (10^9^/L)	0.55 (0.42–0.74)	0.53 (0.38–0.77)	0.665	0.55 (0.41 –0.74)	0.53 (0.36–0.75)	0.463
NE (10^9^/L)	5.29 (3.36–7.00)	5.37 (3.63–7.90)	0.172	5.16 (3.32 –6.98)	4.95 (3.43–7.28)	0.691
TD (μmol/L)	10.4 (8.1–13.8)	10.5 (8.2–13.7)	0.742	10.4 (8.1–13.7)	10.95 (8.85–14.45)	0.153
Cr (μmol/L)	71 (60–84)	71 (59–90)	0.962	72 (60–84)	72 (58–89)	0.725
AST (U/L)	21 (17–27)	21 (17–28)	0.551	21 (17–26)	21 (17–27)	0.546
ALT (U/L)	16 (12–24)	16 (10–27)	0.661	15 (12–24)	16 (10–27)	0.972
DD (μmol/L)	1.9 (1.5–2.9)	2.1 (1.6–2.7)	0.131	1.9 (1.5–2.8)	2.1 (1.7–2.7)	0.050
ID (μmol/L)	8.2 (6.5–11.3)	8.5 (6.5–11.0)	0.918	8.2 (6.5–11.3)	8.9 (6.9–11.5)	0.106
ALB (g/L)	36.4 (33.2–38.9)	34.9 (31.0–38.6)	0.054	36.8 (33.3–38.9)	35.9 (32.6–39.5)	0.863
CK-MB (ng/mL)	10 (2–15)	10 (3–15)	0.420	10 (2–15)	10 (5–16)	0.098
cTnI-HS (ng/mL)			0.190			0.201
>ULN	23 (14.4%)	62 (19.2%)		19 (12.3%)	27 (17.5%)	
≤ULN	137 (85.6%)	261 (80.8%)		135 (87.7%)	127 (82.5%)	
IL-2 (pg/mL)	0.36 (0.01–1.10)	0.39 (0.01–0.91)	0.500	0.39 (0.01–1.11)	0.34 (0.01–0.81)	0.273
IL-4 (pg/mL)	0.35 (0.01–0.67)	0.21 (0.01–0.60)	0.026	0.35 (0.02–0.69)	0.21 (0.01–0.57)	0.014
IL-6 (pg/mL)	74.52 (11.95–215.81)	73.06 (13.56–211.26)	0.715	74.52 (12.08–210.78)	79.95 (12.97–211.80)	0.880
IL-10 (pg/mL)	12.01 (3.42–38.83)	8.02 (2.29–37.99)	0.280	12.61 (3.51–39.18)	7.44 (2.10–46.10)	0.442
TNF-α (pg/mL)	2.09 (0.01–7.34)	0.69 (0.01–5.88)	0.010	2.49 (0.01–7.45)	1.39 (0.01–6.03)	0.053
IFN-γ (pg/mL)	2.45 (0.02–7.23)	1.42 (0.01–7.36)	0.110	2.75 (0.06–7.56)	1.72 (0.01–7.66)	0.228

PSM: propensity score matching; CHD: coronary heart disease; COPD: chronic obstructive pulmonary disease; WBC: white blood cell count; LY: lymphocyte count; Mono: monocyte count; NE: neutrophil count; TD: total bilirubin; Cr: serum creatinine; AST: aspartate transaminase; ALT: alanine aminotransferase; DD: direct bilirubin; ID: indirect bilirubin; ALB: albumin; CK-MB: creatine kinase-MB; cTnI-HS: high-sensitivity cardiac troponin I; ULN: upper limit of normal value; IL: interleukin; TNF: tumor necrosis factor; IFN: interferon. The upper limit of normal value of cTnI-HS in this study is 0.0175 ng/mL.

## Data Availability

The original contributions presented in this study are included in the article/Appendix A. Further inquiries can be directed to the corresponding author.

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
