# Peer review of "Differences in Non-Pathogenic Lung-Colonizing Bacteria Among Patients with Different Types of Pneumonia: A Retrospective Study"

_microorganisms, 2025, doi:10.3390/microorganisms13092099_

Round 1
Reviewer 1 Report
Comments and Suggestions for Authors
I read the article “Differences in Non-Pathogenic Lung-Colonizing Bacteria Among Patients with Different Types of Pneumonia: A Retrospective Study” with interest, and it is relevant to the journal.
Overall, the introduction and objectives are clear, and the discussion is effective. I note only one major limitation, which could be addressed by integrating results on the type of pathogen causative for the pneumonia. These pathogens could in fact act as competitive factors or otherwise alter the non-pathogenic flora, and knowing them would be essential—especially in cases of mixed pneumonias (bacterial and fungal), for which data are not reported. Finally, please specify in the methods section how the costs were defined.
Other minor issues to be addressed:
90: What do authors mean with follow-up? please add the mean hospitalization stay.
Supplementary materials: table ok, figure ok. Clear and representative.
sincerely
Reviewer 2 Report
Comments and Suggestions for Authors
This paper analyzes data from 483 patients with various types of pneumonia who underwent bronchoscopy and mNGS (BALF) diagnostics. The authors focus on the potential impact of nonpathogenic bacteria colonizing the lungs on the course of the disease and prognosis. Some of these bacteria, such as Rothia mucilaginosa and Prevotella melaninogenica, appear to have a beneficial effect on the course of pneumonia, while others, such as Veillonella parvula, may be associated with a more severe course. The paper makes an interesting contribution but requires methodological refinement, greater interpretative caution, and structural corrections.
1. The Blurring of the Concept of "Nonpathogenic Colonizing Bacteria"
The authors classify many bacteria as "nonpathogenic," but in reality, some (Veillonella, Fusobacterium, Streptococcus mitis) are opportunistic and may play a pathogenic role, especially in immunosuppressed patients. It is necessary to clarify the basis for classifying these microorganisms and discuss the limitations of this classification.
2. Overly strong suggestions of a "protective effect" without confirmation of causality
The authors use terms such as "probiotic effect" or "beneficial effect" (e.g., Rothia mucilaginosa), despite the fact that the study cannot demonstrate a causal relationship. It is necessary to soften the wording and introduce precise correlational language.
3. Lack of control for the effect of antibiotic therapy
All patients received antibiotics, which likely significantly influenced the composition of the lung microbiota. The authors should acknowledge this as a major confounding factor and discuss its impact.
4. PSM does not compensate for all differences between groups
Although the authors used propensity score matching, the analysis is limited to a few variables (age, gender, selected comorbidities). It is unknown whether other potentially important factors (e.g., prior hospitalizations, disease severity, type of antibiotics) were included.
5. Opaque criteria for classifying bacteria as "colonizing" rather than "pathogens"
The definition (p. 3) indicates that bacteria considered nonpathogenic were classified by two physicians – a subjective approach, unsupported by clinical standards (e.g., IDSA, CDC). It is recommended to clarify and/or use objective RPM-r thresholds.
6. Overly general analysis of microbiological data
The paper relies on the presence/absence and number of sequences. Information is missing on: relative abundance, co-occurrence, network analysis, relationship to the dominant flora (e.g., was Rothia dominant or merely present?).
7. Lack of biological validation (e.g., cytokine levels in the context of specific bacteria)
Correlations between bacteria and IL-6, TNF-α, and IFN-γ are indicated (p. 9), but these are not supported by multivariate analysis that would account for the influence of pneumonia type or prior treatment.
Reviewer 3 Report
Comments and Suggestions for Authors
In this retrospective study the authors analyzed lung samples from patients with pneumonia and found that certain bacteria (usually those considered as harmless) can influence the course of the disease. Some bacteria (e.g., Rothia mucilaginosa or Prevotella melaninogenica) were associated with better outcomes, suggesting that the lung microbiome could play a role in pneumonia management.
The study and its findings are rather interesting.
Some suggestions for further improvement.
1. Since the study was retrospective, how did you take into consideration the antibiotic treatment ?
(since the antibiotic therapy of each patient may affect the outcome under study)
2. In the same context, how co-morbidities can influence the association between pneumonia and the colonizing flora?
3. The patients in your study (i.e., the retrospective data) came from a Chinese population. Do you think that the findings can be generalized to other ethnicities ?
4. In order to reduce residual confounding you could further apply multivariate regression analysis or other multivariate techniques.
Minor:
5. Some Figures and Tables are rather important, have you considered to place them in the main manuscript (instead in the Suppl. material) ?
Round 2
Reviewer 1 Report
Comments and Suggestions for Authors
much clear and supported.
Reviewer 2 Report
Comments and Suggestions for Authors
The Authors have revised manuscript accordingly to Reviewer's suggestions.
Reviewer 3 Report
Comments and Suggestions for Authors
Thank you for your collaboration.
The authors have addressed my comments. I have no further comments to make.